# VR-Mediated Cognitive Defusion: A Comparative Study for Managing Negative Thoughts

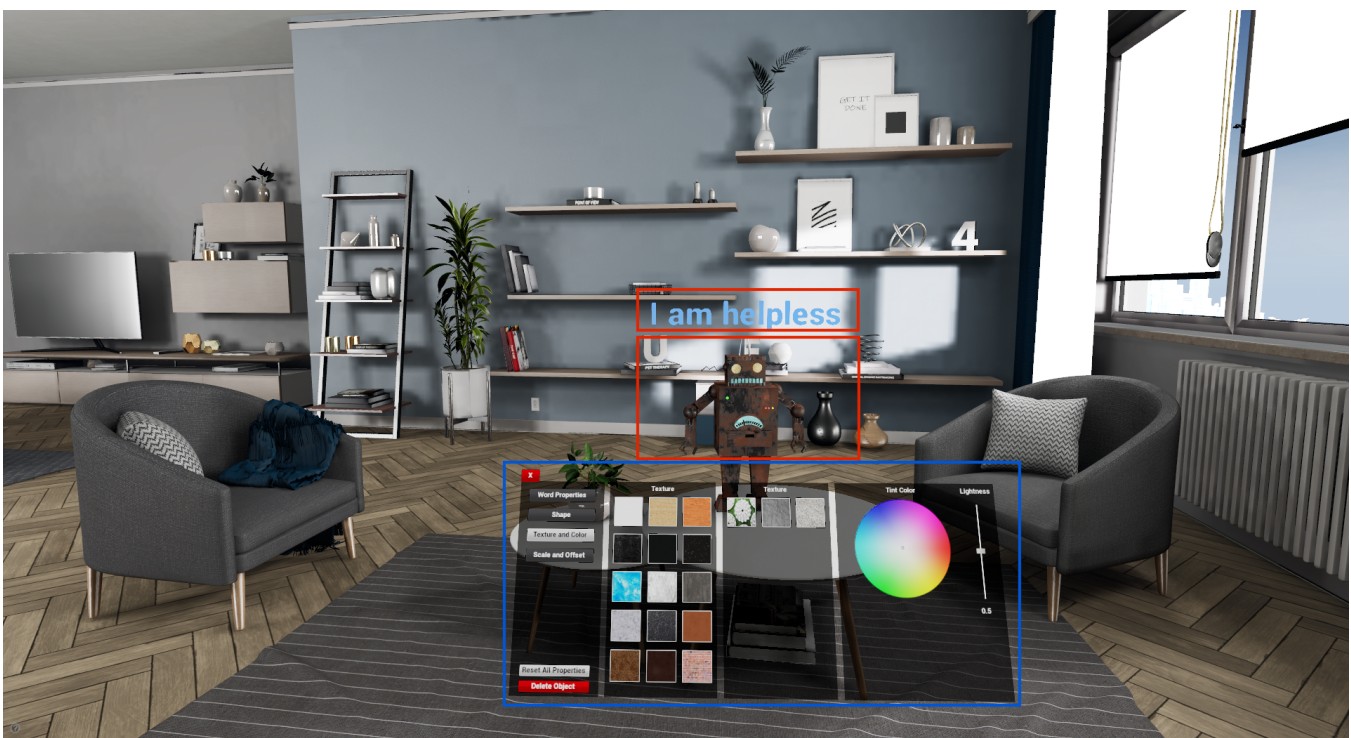

**Figure 1: The VR setup for ACT's cognitive defusion exercise allows users to manipulate a virtual object (colored red) that represents a negative thought. Through the interface menu (colored blue), users can modify the object's shape, texture, color and scale. Additionally, users can grab and reposition the object onto various surfaces, such as tables and shelves.**

## ABSTRACT

The growing prevalence of psychological disorders underscores the critical importance of mental health research in today's society. In psychotherapy, particularly Acceptance and Commitment Therapy (ACT), cognitive exercises employing mental imagery are used to manage negative thoughts. However, the challenge of maintaining vivid imagery diminishes their therapeutic effectiveness. Virtual reality (VR) offers untapped potential for increasing engagement and therapeutic efficacy. However, there is still a gap in exploration regarding how to effectively leverage the potential of VR to enhance traditional cognitive exercises with mental imagery. This study investigates the effective HCI design and the comparative efficacy of a VR-mediated exercise for promoting cognitive defusion to address negative thoughts grounded in ACT. Using a co-design approach with clinicians and potential users of postgraduate students, we developed a VR system that materializes negative thoughts into tangible objects. This allows users to visually modify and transpose these objects onto a surface, facilitating mental detachment from negative thoughts. In an evaluation study with 20 non-clinical participants, divided into VR and mental imagery groups, we assessed the impact of the cognitive defusion exercise on their perception of negative thoughts and psychological measures using standardized questionnaires. Results show improvement in both groups, with significant enhancements in negative thought perception and mental detachment from negative thoughts exclusively in the VR group, whereas the mental imagery group did not demonstrate significant changes. Interviews emphasize the VR's capability to present vivid visualizations of negative thoughts effortlessly, highlighting its effectiveness and engagement in psychotherapy to facilitate cognitive exercises.

## CCS CONCEPTS

• **Human-centered computing → Empirical studies in HCI**.

## KEYWORDS

Human-Computer Interaction, Virtual Reality, Psychotherapy, Cognitive Defusion

## 1 INTRODUCTION

The integration of technology, particularly Virtual Reality (VR), into mental health interventions leverages insights from Human-Computer Interaction (HCI), psychology, and psychotherapy [10][30]. VR's benefits, like minimizing external distractions, providing immersive visual cues, and offering instant feedback, make it a compelling tool for therapeutic applications [7]. Researchers have been increasingly adopting VR to facilitate mental imagery-based cognitive exercises to manage negative thoughts. VR's immersive capabilities and interactivity allow users to visualize, adjust, and interact with therapeutic content effectively [25]. Exiting research by Prudenzi et al. [29] has highlighted VR's potential capability in aiding individuals to confront negative thoughts. In a blank virtual space, users can represent a negative thought as a 3D text object (e.g., "*I am a failure*") against a background plane, and modify its properties such as shape, color, and size. This technique, based on Acceptance and Commitment Therapy (ACT) [15], aims to enable cognitive defusion by helping users understand that negative thoughts are merely mental constructs composed of mental images or words. In VR, users interact with their thoughts as distinct, controllable objects in an immersive setting, thus changing their relationship with these thoughts and lessening their adverse impact.

However, the proof-of-concept study reveals limitations in HCI designs and evaluation. The VR system's inability to visualize negative thoughts as tangible objects, in VR deviates from the original exercise. Additionally, it overlooks an essential task of object transposition (i.e., place an object onto a surface like a table), which Wagener et al. [41] suggest could aid in managing negative thoughts in their "*Wrapping It Up*" design concept. The lack of a structured design methodology, such as co-design, raises questions about the efficacy of VR designs to facilitate the exercise. Moreover, the absence of a direct comparison between VR and mental-imagery interventions in the study's evaluation phase limits the ability to ascertain VR's comparative effectiveness in facilitating cognitive defusion. These limitations underscore the necessity for further exploration and refinement in both the design and assessment of the VR-based exercise.

To address this knowledge gap, our study aims to: 1) develop HCI designs for the VR-based cognitive defusion exercise, and 2) compare the efficacy with conventional mental-imagery format as shown in Figure 2. Our design study utilized a co-design approach, involving mental health experts and postgraduate students as potential users, to design and develop a VR system for the exercise. The VR system allows users to embody their negative thoughts as tangible objects within a virtual living room filled with everyday objects, and change the visual properties like color, texture, and size. Moreover, users can place these objects on surfaces like a table and shelf, addressing a task not explored in the prior research. We tested the efficacy through a randomized control trial with 20 postgraduate students, comparing VR to a mental-imagery intervention. While both groups showed improvements in psychological measures through standardized questionnaires, only the VR group reported a statistically significant enhancement in perceptions towards negative thoughts and scores on the State Cognitive Fusion Questionnaire (SCFQ) [2]. This difference can be attributed to VR's immersive environment, as noted in participant interviews, highlighting VR's superior efficacy over mental-imagery for managing negative thoughts.

Our contributions include a co-designed VR system for the defusion exercise in ACT, offers insights for integrating psychological techniques with VR to deal with negative thoughts, and demonstrates VR's superior efficacy over mental imagery through a comparative evaluation.

## 2 RELATED WORK

This section introduces studies on 1) design dimensions centered on visualizing emotions, 2) therapeutic techniques, particularly ACT, and 3) VR for cognitive defusion, setting the context for our study.

### 2.1 Design Approaches for Visualizing Emotional Experiences

Several studies explored visualizing emotions, thoughts, or past challenges in VR as a method for enhancing psychological well-being, utilizing immersive visualization techniques [21]. Applications like Mood Worlds [39] and SelVReflect [40], inspired by art therapy, demonstrate VR's potential in inducing positive emotions and self-reflection by letting users freely visualize emotions and past challenges in VR. A similar VR application, Emotion Brush by Kitson et al. [20], incorporated a cognitive reappraisal tactic to further encourage users to reinterpret their negative thoughts by repainting the virtual world into something more positive. While these existing methods highlight autonomous emotional expression in virtual worlds to evoke positive emotions, our research investigates VR's capabilities to support a psychotherapy-based therapeutic technique. This technique originally employs mental-imagery to alter and reposition visualized negative thoughts to view them from a new perspective, aiding in mental detachment from those. We are motivated to explore how immersive interaction and environments can enhance these therapeutic tasks in a vivid and intuitive way, potentially offering significant improvements over traditional mental-imagery format.

### 2.2 Technological Adoptions in ACT

This study explores the designs and efficacy of a VR-mediated cognitive defusion exercise within Acceptance and Commitment Therapy (ACT) [15], a mindfulness-based third-wave psychotherapy, focusing on "destructive normality" — the natural occurrence of unhelpful thoughts and feelings [14]. Distinguished from traditional therapy methods that aim to directly alter dysfunctional thinking or remove distress, ACT enhances psychological flexibility, teaching detachment and acceptance of thoughts as part of the human experience to live a value-aligned life. To support cognitive exercises provided in ACT, digital technologies are increasingly used [19], including videoconferencing [43], web-based components [27], and apps for anxiety and chronic pain [17] [12],

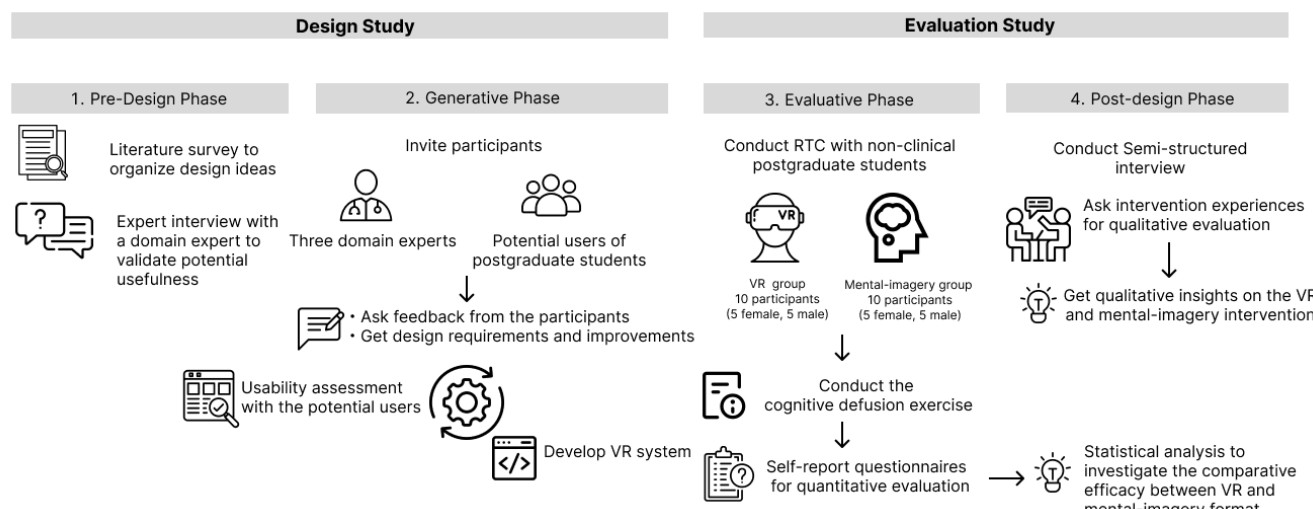

**Figure 2: Overview of the Four-Phase Study Design and Evaluation Framework. 1. The pre-design phase includes a literature survey and expert interviews to lay the groundwork. 2. The Generative Phase involves participant recruitment and feedback collection to refine the VR system. 3. The Evaluative Phase includes a Randomized Controlled Trial with non-clinical postgraduate students comparing VR and mental-imagery interventions. 4. The Post-Design Phase entails semi-structured interviews to extract qualitative data and statistical analysis to assess the efficacy of both formats in managing negative thoughts.**

showing efficacy similar to face-to-face interventions [23]. Virtual Reality Exposure Therapy (VRET) [8] has emerged as a valuable tool in ACT, offering controlled environments for overcoming fears and avoidance behaviors, with positive outcomes for conditions like social anxiety, acrophobia, and alcohol use disorder [13] [4] [5]. While VRET primarily immerses users in virtual environments, this study extends beyond immersion by enabling users to interact with virtual objects to facilitate a cognitive defusion exercise.

## 2.3 VR for Cognitive Defusion

Cognitive defusion is one of the key therapeutic processes in ACT [15], primarily aiming to help clients detach from negative thoughts by perceiving negative thoughts as mere images or words. This facilitates a shift in perspective of negative thoughts and reduces their detrimental influence on thoughts and behaviors. Among various cognitive exercises for promoting cognitive defusion, we focus on an exercise that originally employs mental-imagery, which is a kind of cognitive restructuring [25]. This exercise involves observing a negative thought as a tangible object, describing it, altering its properties, and transposing it on a surface within a virtual space [15], as outlined in supplementary material (A).

Prudenzi et al. [29] explored the potential efficacy of VR for this exercise, focusing on visualizing a negative thought as the corresponding text object (e.g., "*I am a failure*") against a background plane, and modifying the object's properties such as its shape, size, color in a blank virtual environment. The user study with university students showed improvements in managing negative thoughts and an increased state of detachment from the thoughts. However, the proof-of-concept study reveals limitations in HCI designs and evaluation of the VR-mediated exercise, notably its inability to visualize negative thoughts as tangible objects and to incorporate the

crucial task of placing objects on surfaces. The absence of a structured design methodology and the lack of a comparative evaluation with conventional mental imagery format leave the efficacy of VR exercise uncertain.

Our research aims to explore the effective HCI designs through co-design and compare the VR exercise with traditional mental imagery format. We are particularly interested in the "Wrapping It Up" concept [41], which uses object placement of symbolized negative thoughts for a perspective change toward them, addressing a gap in the prior study. Inspired by Paul et al.'s suggestions [11], we aim to explore VR's unique features and differences in psychological interventions compared with mental-imagery format. We hope this comparative analysis will guide future clinical research and the integration of VR defusion exercises into mental health treatments.

## 3 VR SYSTEM DESIGN

This section outlines the design methodology, design considerations and design features behind our VR system to facilitate the cognitive defusion exercise.

## 3.1 Design Methodology

We employed a co-design methodology [31] [28], collaborating with three domain experts, including a clinical psychologist, a school counselor, and a psychiatrist, who have expertise in ACT. In terms of prospective users, we invited postgraduate students from our local university to participate in the study. The rationale behind this selection stemmed from the recognition that university students often contend with significant academic pressures, rendering them susceptible to mental health issues [36]. Our approach adhered to the well-regarded framework proposed by Sanders and Stappers [32], which deconstructs the design journey into four

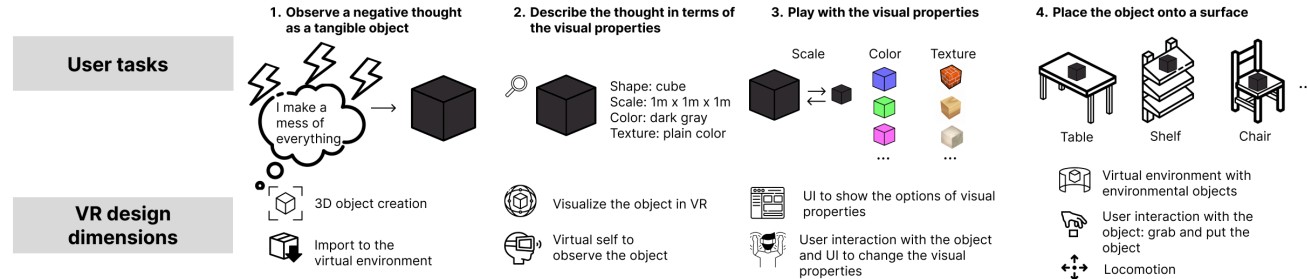

**Figure 3: User tasks and design dimensions for the cognitive defusion exercise**

interlinked phases: 1) Pre-design Phase 2) Generative Phase 3) Evaluative Phase and 4) Post-design Phase as mentioned and illustrated in Figure 2.

## 3.2 Design Considerations

Throughout the generative phase, we organized the design considerations into four integral components guided by the taxonomy for the dimensional elements of XR prototypes proposed by Krauss et al [22]. These four components encompass: *virtual self*, 2) *virtual environment*, 3) *content*, and 4) *user interaction*.

In our VR system, we focused on two key design considerations for the *virtual self*, which is the user's embodiment in the virtual environment: **(D1.1) enhancing presence and immersion through virtual self** and **(D1.2) representation and interaction via virtual hands**. We aimed to bolster presence and immersion, acknowledging the virtual self's importance in creating self-location, agency, and ownership, essential in VR therapeutic exercises as highlighted by Slater et al. [35] and Lindner et al. [25]. Additionally, we emphasized the representation of hands in the VR environment, crucial for interaction with objects in the defusion exercise [6], while also considering user preferences for the virtual self's appearance, balancing gender diversity and comfort with its visual realism.

To create an effective *virtual environment*, two key design considerations are essential: **(D2.1) aligning with the defusion exercise's goal** of enabling users to detach from negative thoughts through visualization and manipulation of objects, and **(D2.2) ensuring relevance to everyday life** for intuitive user interaction. Unlike mindfulness exercises that often use natural settings for relaxation [21], our focus is on activities that mimic daily experiences, enhancing the exercise's practicality and user engagement.

Regarding the *content*, the virtual environment should create an immersive setting with **(D3.1) environmental elements like furniture and decor to link negative thoughts with familiar surroundings**, facilitating cognitive defusion. **(D3.2) Presenting negative thoughts as tangible objects with the corresponding text** according to ACT's externalization principle. Furthermore, **(D3.3) an interface to customize these objects' visual properties** allows users to interact with the tangible objects. Lastly, **(D3.4) a system-wide interface for system setting** enhances usability, catering to diverse preferences and ensuring a customized VR experience.

As for *user interaction*, Figure 3 illustrates how each user task in the cognitive defusion exercise corresponds to specific design dimensions within the virtual environment. Especially, **(D4) object manipulation for negative thought representation** involves two primary interactions: object selection and manipulation to change the visual properties and location. These require virtual hands to interact with objects and user interfaces to select available options for visual changes.

## 3.3 Design Features

To enhance user immersion, our VR system transitioned from virtual hands to full arms with inverse kinematics for synchronized movements, improving user experience **(D1.1)**. Considering the virtual self's appearance **(D1.2)**, user feedback favored non-realistic robotic models over realistic ones to reduce discomfort from mismatches between their physical and virtual ones. Supported by studies indicating visual realism's limited effect on virtual body ownership [34], we adopted robotic hands and arms as the standard virtual representation, offering an option for realistic models based on user preference.

Our VR system features a virtual living room, chosen for its relevance to the cognitive defusion exercise **(D2.1)** and everyday familiarity **(D2.2)**. This setting, adapted from an Epic Games sample [1], is designed to support the exercise's objective of thought detachment by providing surfaces for object transposition and environmental context that allows users to see their representations of negative thoughts from different perspectives (see Figure. 1).

The virtual living room features static objects for a realistic ambiance, with certain items such as a table, chair and ornaments, especially for object transposition **(D3.1)**. Negative thoughts are visualized as pre-imported objects that were associated by users with adjustable visual properties like shape, texture, color and size, combined with descriptive text **(D3.2)**. An object menu enables users to change these properties, supporting cognitive defusion by facilitating the externalization and manipulation of negative thoughts **(D3.3)**. Additionally, a system settings menu in the VR environment allows for customization of hand type, navigation, and object management, enhancing user experience **(D3.4)**.

To support user interaction **(D4)**, we incorporate a ray-casting laser pointer for object selection and choosing options on the object menu for visual changes, providing audio feedback for user

---

[1]https://docs.unrealengine.com/4.26/en-US/Resources/Showcases/ArchVisInterior/

confirmation. For object transposition, users can use virtual hands to grab, move, and place objects onto surfaces in VR.

## 4 EVALUATION

The evaluation used a $2 \times 2$ mixed experimental design, comparing VR (intervention) and mental-imagery (control) groups across pre-post assessments. Dependent variables were psychological measure scores of self-report questionnaires. Our hypothesis suggests significant within-group differences in psychological measure scores before and after interventions, expecting higher efficacy in the VR group compared to the mental-imagery group. The evaluation followed our university's human research ethics protocol and received ethical approval. Participants were compensated with HKD 100 for their involvement.

### 4.1 Participants

Our study targeted postgraduate students, a group often facing significant academic stress and mental health challenges, as indicated by existing research [9]. We recruited participants by convenience sampling from our university who were 18 years or older, had normal vision, no history of epilepsy, and no clinical mental health disorders in the past six months. Using the Beck Depression Inventory-II [1] for screening, we excluded those with severe (scores above 30) or minimal depression (scores below 10). Our final sample included 20 postgraduate students (10 males and 10 females). Table 1 details their gender, condition, target negative thoughts, and associated objects, with further explanation on these selections provided later.

### 4.2 Psychological Measures

We collected three types of data through: 1) target thought measures, 2) self-report questionnaires, and 3) qualitative feedback on the intervention through semi-structured interview.

*4.2.1 Target Thought Measure.* Participants rated a target negative thought on a 1-10 Likert scale before and after the intervention, following methodologies from previous studies [24] [29]. A score of 1 signifies the thought is perceived as extremely negative, believable, uncomfortable, and unwelcome, while a score of 10 represents the thought as positive, unbelievable, comfortable, and acceptable. We anticipate significant post-intervention rating increases in both groups, with a greater increase in the VR group.

*4.2.2 Self-report Questionnaires.* We administered three self-report questionnaires before and after the intervention: 1) State Trait Anxiety Inventory (STAI) [37], 2) State Cognitive Fusion Questionnaire (SCFQ) [2] and 3) The Acceptance and Action Questionnaire–II (AAQ-II) [38]. These questionnaires were selected based on their established validity and reliability in measuring various aspects of mental health and well-being. We hypothesize that the scores for these measures will significantly decrease after the intervention in both groups, with a more decrease in the VR group.

*4.2.3 Qualitative Feedback.* We collected qualitative feedback after the intervention in both groups through semi-structured interviews, each lasting 15-30 minutes. The interview was guided by questions listed in supplementary material (B) and included additional questions to capture participants' experiences accordingly.

### 4.3 Procedure

Participants were individually invited into a quiet room for a briefing on the study and its tasks. After a briefing on the experiment procedures and consent forms, we collected the signatures for consent from those willing to participate. Participants were then randomly assigned to either the VR group (five males and five females) or the mental-imagery group (five males and five females) to proceed with the designated tasks as below.

**Tutorial session of the VR system:** For the VR group, we provided a demonstration of using the VR system for the defusion exercise, covering navigation in the virtual environment, menu interaction, and how to create, manipulate, and transpose objects. Participants were then invited to familiarize themselves with the VR environment and system until they felt comfortable.

**1. Selection of the target negative thought:** Participants from both groups were asked to choose a target negative thought from a list of 10 common negative thoughts in supplementary material (C), such as "*I am a failure*" and "*My life is pointless*", derived from a study by Healy et al. [16]. To ensure the thought was negatively perceived, participants rated it on negativity, believability, discomfort, and unwillingness to have the thought on a scale from 1 (most negative) to 10 (most positive). Only thoughts rated as 1 or 2 were selected for use; higher-rated thoughts required selecting another from the list. In the VR group, participants were further asked to identify an object to represent the selected negative thought. They provided a drawing or image of this object, which we used to create a 3D model. Upon their confirmation, we imported the models into the VR system for their intervention.

**2. Autobiographical Emotional Memory Task (AMET):** We utilized Autobiographical Emotional Memory Task (AEMT) [26], a mood induction procedure (MIP) designed to evoke specific moods in a controlled setting [42] before the intervention. AEMT, known for effectively inducing positive and negative moods [18], required participants to recall and write down memories related to their target negative thoughts for 10 minutes. This exercise aimed to ensure participants were in the relevant emotional state for the defusion exercise. To protect privacy, the written content were not collected and reviewed by the evaluator.

**3. Pre-assessment:** Participants filled in an online form, rating their negative thoughts on negativity, believability, comfort, and willingness on a 1-10 scale and answering three questionnaires: STAI, SCFQ, and AAQ-II to evaluate their mental state before the intervention.

**4. Intervention:** Participants performed the defusion exercise for 10 minutes, using either their mental imagery or our VR system, depending on their group assignment.

In the mental-imagery group, participants either closed or opened their eyes as preferred and followed defusion exercise instructions. Each participant visualized their selected negative thought as a tangible object, mentally modifying its color, texture, shape, or size. They then imagined placing this object on a chosen surface, such as a chair, table, or shelf, to observe it.

**Table 1: Overview of the participants in the experiment**

| P | Age | Gender | Condition | Target negative thought | An object to represent the thought |
|---|-----|--------|-----------|--------------------------|-------------------------------------|
| P1 | 24 | Male | Mental-Imagery | No one will ever love me | A pile of sticky mud |
| P2 | 26 | Male | Mental-Imagery | I am a failure | A bad character in a comic boook |
| P3 | 29 | Male | Mental-Imagery | I am helpless | Black untouchable cloud |
| P4 | 27 | Female | Mental-Imagery | I am helpless | A fishbowl with a fish |
| P5 | 26 | Female | Mental-Imagery | I make a mess of everything | A black cube |
| P6 | 28 | Female | Mental-Imagery | Sometimes I wish I wasn't me | Many bubbles with different scales |
| P7 | 23 | Male | Mental-Imagery | I make a mess of everything | Many brown and green leaves |
| P8 | 28 | Female | Mental-Imagery | My life is pointless | A large, hard and dirty stone |
| P9 | 27 | Male | Mental-Imagery | I am a bad person | Dark cloud |
| P10 | 25 | Female | Mental-Imagery | I am stupid | Many rolled papers |
| P11 | 27 | Male | VR | My life is pointless | A bumpy and broken meteorite |
| P12 | 32 | Male | VR | I make a mess of everything | Splashes of colorful paints |
| P13 | 28 | Male | VR | No-one will ever love me | A regular dodecahedron |
| P14 | 26 | Male | VR | I am failure | A rusty robot |
| P15 | 27 | Female | VR | My life is pointless | A cliff |
| P16 | 26 | Female | VR | I make a mess of everything | Tangled gray wires |
| P17 | 32 | Male | VR | I make a mess of everything | A cubic monster with eyes |
| P18 | 26 | Female | VR | I am broken | A man burying his head in his hands |
| P19 | 26 | Female | VR | Sometimes I wish I wasn't me | A cube with wrinkled texture |
| P20 | 25 | Female | VR | My life is pointless | A double-sized bed |

In the VR group, participants donned a Meta Quest2 headset, adjusting the lens and strap for optimal clarity. They completed the same defusion exercise as the mental-imagery group, initially selecting and placing a virtual object that symbolized their negative thought within the environment. Participants then modified the object's appearance using the object menu. Finally, they positioned the object on a chosen surface within the virtual living room to observe it. The VR session was displayed on a laptop for guidance and recorded for review. All participants in both groups completed the tasks within the designated time frame.

**5. Post-assessment:** Participants filled in the same online form as the post-assessment that consists of the target thought measure and three questionnaires (STAI, SCFQ and AAQ-II).

**6. Semi-structured interview:** We conducted semi-structured interviews with participants from both groups to collect qualitative feedback, using questions in supplementary material (B) and additional prompts for more insights accordingly. The conversations were summarized and compiled into a document available in supplementary material (D).

### 4.4 Result

Table 2 shows the average scores and standard deviations for the Target Thought Measure (TTM), STAI, SCFQ, and AAQ-II before and after the interventions for each group. We analyzed these measures' mean differences pre- and post-experiment, illustrated in Figure 4, using a mixed $2 \times 2$ ANOVA for time × group (VR/Control). Qualitative findings from semi-structured interviews are also provided to supplement the quantitative data.

*4.4.1  Target Thought Measure.* The Target Thought Measure showed an average improvement of 2.8 points in the VR group and 2.3

points in the control group. ANOVA results ($F(1, 18) = 0.157$, $p = 0.694$) indicated no significant time influence difference between groups. However, the VR group exhibited a significant pre-post increase ($F(1, 18) = 9.408$, $p = 0.006$), unlike the control group ($F(1, 18) = 6.991$, $p = 0.016$), which showed no significant change.

*4.4.2  Self-report Subjective Measures.* For STAI, the VR group saw a 20-point decrease and the control group an 18.3-point decrease in average at post-experiment. ANOVA results ($F(1, 18) = 0.076$, $p = 0.783$) showed no significant time influence difference between groups. Nonetheless, both groups experienced significant reductions in STAI scores, with the VR group ($F(1, 18) = 21.025$, $p = 0.0002$) and the control group ($F(1, 18) = 18.020$, $p = 0.0004$) showing significant improvements.

For SCFQ, both the VR and control groups saw an average decrease of 13.3 and 12.9 points, respectively. ANOVA results ($F(1, 18) = 0.004$, $p = 0.949$) indicated no significant difference in time influence between the groups. Yet, the VR group showed a significant reduction in SCFQ scores ($F(1, 18) = 10.188$, $p = 0.005$), unlike the control group, which showed no significant change ($F(1, 18) = 7.357$, $p = 0.014$).

For AAQ-II, the VR group's scores decreased by 10.9 points and the control group's by 10.2 points in average at post-experiment. ANOVA results ($F(1, 18) = 0.019$, $p = 0.888$) showed no significant time influence difference between groups. The VR group saw no significant change in AAQ-II scores ($F(1, 18) = 7.082$, $p = 0.015$), while the control group experienced a significant reduction ($F(1, 18) = 12.990$, $p = 0.002$).

*4.4.3  Semi-structured Interview.* Participants visualized their negative thoughts as various objects like sticky mud, clouds, and cubes without a consistent pattern, even for identical thoughts **(Q1.1).**

**Table 2: Descriptive statistics on the pre-post psychological measures for mental-imagery (MI) and VR group**

|  | TTM | | STAI | | SCFQ | | AAQ-II | |
| --- | --- | --- | --- | --- | --- | --- | --- | --- |
|  | M | SD | M | SD | M | SD | M | SD |
| **Pre** | | | | | | | | |
| MI | 2.7 | 1.82 | 55.1 | 11.26 | 32.7 | 11.37 | 28.6 | 5.60 |
| VR | 2.3 | 0.94 | 57.7 | 11.64 | 33.1 | 10.55 | 32.2 | 11.29 |
| **Post** | | | | | | | | |
| MI | 5.0 | 2.05 | 36.8 | 7.67 | 19.8 | 9.84 | 18.4 | 6.97 |
| VR | 5.1 | 2.72 | 37.7 | 7.39 | 19.8 | 7.88 | 21.3 | 6.34 |

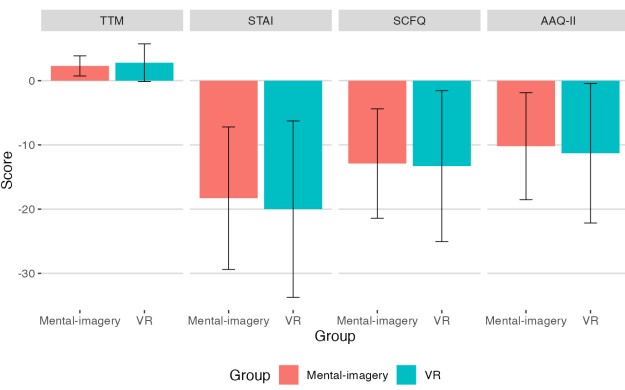

**Figure 4: The average of pre-post changes in the score of the measures by group**

They related negative thoughts to different shapes, colors, textures, and sizes **(Q1.2)**. For instance, angular shapes were linked to hurt, while tangled wires suggested chaos. Dark colors often symbolized negative moods, whereas light colors could indicate positive sensations or emptiness, with mixed colors representing complex negativity. Textures like dirty stone conveyed aversion, and size reflected the intensity of negativity, with larger objects denoting stronger negative feelings. These descriptions show personalized associations between thoughts and their visual representations.

The importance of each visual property for representing negative thoughts was rated on a 1-5 scale, with shape being the most crucial (Avg = 4.0, SD = 1.21), followed by color (Avg = 3.8, SD = 1.2), scale (Avg = 3.7, SD = 1.4), and texture (Avg = 3.4, SD = 1.5). Shape's significance aligns with ACT's view that thoughts involve specific shapes, while color's importance reflects its role in expressing moods **(Q3.1)**. Participants found reducing size and brightening colors effective for distancing negative thoughts, linking negativity to these properties. While shape alterations were less favored to preserve the metaphor, changes in motion and quantity also aided cognitive defusion **(Q3.2)**. Placing objects on a surface was seen as beneficial for creating mental distance and acceptance, validating the effectiveness of the "Wrapping It Up" design element [41] **(Q3.3)**.

Participants rated the vividness and expressiveness of their negative thought representations on a 1-5 scale, showing no significant

difference between groups but a slightly higher average in the VR group (mental-imagery: Avg 4.0, SD: 1; VR: Avg 4.3, SD: 0.71) **(Q2)**. Some participants in only mental-imagery group reported difficulties in visualizing and transposing objects and staying focused due to distracting surroundings **(Q4)**.

## 5 DISCUSSION

We discuss the effectiveness of the cognitive defusion exercise for managing negative thoughts in VR and mental-imagery groups, assessing both within-group and between-group efficacy. Additionally, we discuss design implications for future clinical interventions.

### 5.1 Statistically Significant Pre-post Effects in Both Groups

The analysis showed significant pre-post changes in some psychological measures for both VR and mental-imagery groups, with VR participants seeing improvements in Target Thought Measure, STAI, and SCFQ, but not AAQ-II, and mental-imagery participants improving in STAI and AAQ-II. These outcomes reflect the effectiveness of ACT-based defusion exercises in both modalities. Participants' feedback in the semi-structured interviews from both groups highlighted the positive impact of the exercise on managing negative thoughts. VR participants enjoyed a vivid and engaging experience, finding the virtual interaction intuitive for observing changes. Mental-imagery participants appreciated the exercise for providing mental distance from negative thoughts, noting the ease of following steps and instructions.

They visualized symbolic objects to represent negative thoughts, with rationales based on visual attributes like shape, color, texture, and scale. Despite variability in shape selection for similar thoughts, patterns in color and texture usage were more consistent, with dark hues and rough textures linked to negative emotions and brighter colors to positive moods. Shape was deemed the most critical property for representing thoughts, supporting ACT's view of thoughts as mental constructions involving specific shapes. Color followed in importance, reflecting its role in expressing moods, as supported by color psychology research. Texture and scale were also important but less emphasized compared to shape and color. This feedback highlights the interventions' alignment with participants' personal experiences, underlining their therapeutic effectiveness.

During the manipulation task, some participants altered these objects' visual properties, noting that smaller object sizes reduced the degree of negativity, illustrating a mental distancing from negative thoughts. Brighter color and smoother texture changes facilitated shifts toward more positive perceptions. However, shape adjustments were less common, due to its importance in maintaining the object's identity and its associated thought.

In the final exercise phase, participants positioned their symbolic objects on surfaces like shelves or chairs. One participant's experience of moving the object from a higher to a lower surface highlighted a change in emotional response—feeling overwhelmed on a high shelf versus less stressed on a lower chair. This act of repositioning helped most participants feel more in control of their thoughts and more accepting of their negativity. The "Wrapping It Up" design element [41], aimed at fostering acceptance and control, proved effective, enhancing the therapeutic benefits of the exercise.

Overall, the steps of representing, manipulating, and repositioning objects aided participants in diminishing the negativity associated with their thoughts, leading to cognitive defusion.

## 5.2 Not Statistically Significant but Higher Effects in VR Group

Our study found no significant difference in the effectiveness between the VR and mental-imagery groups, although the VR group showed a higher efficacy trend. This aligns with a prior study [29] where the VR defusion exercise group saw significant SCFQ score reductions, unlike the control VR group without the exercise. Participant feedback from both groups could shed light on these results.

A potential rationale for the lower performance in the mental-imagery condition can be drawn from challenges participants faced in maintaining mental-imagery to interact with the representations of negative thoughts, reflecting known complexities of mental imagery [3]. While initially able to visualize negative thoughts as objects, continuously imagining and manipulating these objects was difficult. Discrepancies between the physical environment and their mental constructs also caused distractions, hindering full engagement in the exercise. These internal and external factors made focusing on the exercise challenging for some participants.

Conversely, participants in the VR group did not face the challenges reported by the mental-imagery group. The constant presence of symbolic objects in the VR environment eliminated the need for sustained mental imagery, reducing cognitive load as participants could easily observe changes in visual properties, enhancing interaction as noted in previous research [25]. The immersive VR setting effectively isolated them from external distractions, providing a focused environment for the exercises, with the virtual living room offering a calm and conducive setting. Specific interactions, like placing objects alongside everyday items or comparing them to the expansive outside world through a window as depicted in supplementary material (E1 and E2), helped participants achieve mental distance and acceptance of their negative thoughts. These VR-exclusive experiences likely contributed to the observed higher effectiveness in the VR group compared to the mental-imagery group.

## 5.3 Design Implications for Clinical Interventions

The VR designs effectively facilitated the defusion exercise, particularly in visualizing and projecting negative thoughts onto virtual surfaces as anticipated. However, despite satisfying design specifications, participant feedback highlighted potential enhancements beyond our current VR system's capabilities. First, incorporating additional properties such as audio and kinesthetic elements, to depict negative thoughts could more closely match users' mental images of their negative thoughts, providing a broader range of perspectives for addressing them through the property changes. For instance, one participant wished to have her object float in the air, symbolizing its unstable mood, while another sought to emit voice from his object (a rusty robot), bringing it to life. Additionally, customizing the virtual environment could create a more fitting setting for the exercise, extending beyond mere adjustments of furniture and lighting to encompass the entire virtual world. For

instance, one participant chose a cliff to symbolize her negative thought, which did not integrate well with the living room environment we had set up. Offering a variety of virtual scenarios, both indoor and outdoor, like natural landscapes, tailored to the objects representing negative thoughts, would enhance the experience.

Implementing these design features for more adaptable user interaction holds significant potential for clinical users experiencing mental distress, who may have varied needs in managing negative thoughts [44]. Such flexibility not only empowers users to confront and manipulate their negative thoughts in a safe, controlled environment but also accommodates a wider range of negative thoughts. Ultimately, enhancing user interaction in these ways could lead to more meaningful and impactful clinical outcomes, making VR an even more versatile tool to facilitate cognitive exercises.

## 6 LIMITATION AND FUTURE WORK

The workflow for creating 3D models of symbolic objects is manual, posing a challenge for flexible user experience in a self-practice setting. Future research might automate this with generative AI, such as OpenAI's Shape-E [2], while managing the risk of distressing outputs, or empower users with content creation autonomy, as seen in Mood Worlds [39] and SelVReflect [40].

Evaluation-wise, our analysis showed a non-significant but slightly higher effectiveness of interventions among female participants in both groups, potentially linked to their sense of presence and realism during the interventions. Gender differences in presence, often measured by tools like the Igroup Presence Questionnaire (IPQ) [33], have been widely studied with mixed results. Some research suggests men experience higher levels of presence, while other studies find women report greater presence or no significant gender differences. Our interviews did not reveal significant insights on gender impact, suggesting a need for further research with a larger sample to thoroughly investigate gender effects on intervention effectiveness. Besides, the study's focus on a non-clinical, postgraduate student population leaves open the question of VR's effectiveness in clinical settings. Future research should involve clinical patients and consider long-term impacts, moving beyond self-report measures to include behavioral changes and follow-up assessments.

## 7 CONCLUSION

This study explored the HCI designs and comparative effectiveness of a VR-mediated cognitive defusion exercise in ACT, focusing on detaching individuals from negative thoughts and altering their perception of distress. With 20 participants in a gender-balanced, randomized controlled trial, we compared VR intervention against mental imagery format. Both groups showed significant improvements in psychological measures through self-report questionnaires, with a marginally better outcome in the VR group. Interviews provided qualitative insights into VR's advantages for facilitating the exercise in an immersive virtual environment. Despite the lack of a statistically significant difference between interventions, our findings highlight VR's promising potential in ACT for managing negative thoughts, suggesting directions for future research.

---

[2]https://github.com/openai/shap-e

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
