# OpenReview forum: "VR-Mediated Cognitive Defusion: A Comparative Study for Managing Negative Thoughts"
_acmmm.org/ACMMM/2024/Conference — MM2024 Oral_

### Official Review · Reviewer_hPAQ · 2024-05-12

**Rating:** 4
**Confidence:** 1

**Summary:**

In this paper, the authors introduce a framework for enhancing negative thoughts using VR technology. In this framework, the users can play with the visual properties and do some easy tasks, such as placing the object onto a surface.

**Strengths:**

- This work provides an interesting HCI experience based on VR devices.
- This paper is well organized and the figures (such as Fig.2, Fig.3.) are clear enough.
- The experimental results and discussion are illustrated in detail.

**Limitations:**

- The novelty of this study appears to be lacking, as I did not discern any discernible differentiation between this endeavor and contemporary VR-based gaming trends. VR gaming experiences have the potential to ameliorate stress levels among the youth demographic.

**Suitability:**

2

---

### Official Review · Reviewer_SJ2o · 2024-05-20

**Rating:** 5
**Confidence:** 4

**Summary:**

The paper presents a new VR based application, developed based on Acceptance and Commitment Therapy for handling negative thoughts. The application is evaluated with a sample of 20 participants, with encouraging results.

**Strengths:**

(Broadly) well designed and conducted study with (broadly) plausible outcomes
Well-written paper, with a coherent narrative.

**Limitations:**

As a research paper submitted to the research track of ACM MM, the expectation is that there is a research imperative to undertake the work - not just the novelty of yet another therapeutic intervention afforded by VR. The authors mention three contributions their work makes - I would encourage them to use the last "VR’s superior efficacy over mental imagery through a comparative evaluation" to elaborate a research question underpinning the work - and, of course, to answer it as a result of the study presented (which they do).

For transparency and reproducibility, please present full details of the equipment/set up through which the developed VR system was evaluated.

The sample size is small/needs justification for a 2x2 design. It is also unclear how participants were recruited or sampled (more about this later), or indeed if ethics approval was sought from the relevant bodies before undertaking the study.

About the sampling strategy - whilst convenience/purposive sampling with a sample of targeted postgraduate students might have been handy, negative mental thoughts are not the exclusive/main preserve of 24-32 year olds and a wider/broader selection of users should have evaluated the app (middle-aged men have also been shown to have negative thoughts, leading to awful suicide rates in this demographic, for instance). This has to be acknowledged as a limitation of the study.

Also, it is unclear what the familiarity of users with VR is - if all users were recruited from a Computer Science/Design department possibly this had an impact on the complexion of the results.

**Suitability:**

2

---

### Official Review · Reviewer_jRvV · 2024-05-26

**Rating:** 5
**Confidence:** 2

**Summary:**

This paper presents a system for VR-based manifestation of negative thoughts to be used in the cognitive defusion therapy. The system is compared to a mental imagery approach via user study and it is found that while no statistical difference exists VR did seem to perform better at certain metrics.

**Strengths:**

Well structured paper, uses relevant metrics and questionnaires, very detailed, good graphical representation in most of figures, easy to read.

**Limitations:**

For limitation I would say the one the users of the study found themselves. I would say that creation part of the object which represents bad thoughts should have been developed in more detail. The user study is also relatively small.

**Suitability:**

3

---

### Official Review · Reviewer_i1AT · 2024-05-26

**Rating:** 5
**Confidence:** 2

**Summary:**

This paper presents the design of a VR application which serves as an interactive immersive tool for performing a cognitive exercise which aims to address managing negative thoughts. The application allows users to turn their negative thoughts into tangible objects to be modified and repositioned, fostering mental detachment from the thoughts. The solution was evaluated via a user study (N = 20), and its effectiveness was tested against a scenario in which the same exercise was conducted only based on mental imagery.

**Strengths:**

The paper is well-written and utilizes the capabilites of the VR platform - its vividness and immersive qualities - in a very creative way. The overview of existing literature is very informative, and the methodology is presented very clearly, supported by useful figures such as Fig 2 and informative supplementary material. The study methodology seems well designed and executed. I appreciate the thoughtful design of the application and the careful consideration of different aspects of application and environment design, as well as avatar design, as evidenced from the choice of robot hands rather than humanoid hands. The results of this research are certainly promising.

**Limitations:**

The biggest practical issue with this implementation is in the creation of virtual objects. Each 3D model was individually hand-modelled based on the description/sketch provided by the participants. Obviously, this is time consuming and highly dependant on the artistic and technical skills of the person creating these models, as well as highly impractical in the context of personal use or clinical practice. While the authors obviously recognize this concern and suggest concrete solutions, such as the use of generative AI, I would still appreciate if this aspect was discussed in further detail. For example, how long did it take to create these models, is there a possibility that the overall results could be affected by their aesthetic attractiveness, level of detail/realism and the level to which they correspond with what the user had in mind? Even though some of these models can be seen in supplementary material, perhaps adding some examples as images could benefit the manuscript.

**Suitability:**

3

---

### Meta-Review · Area_Chair_9cmA · 2024-07-02

**Recommendation:** Accept (Oral)
**Confidence:** 4

**Metareview:**

The paper presents a VR application designed as an interactive tool for cognitive exercises to manage negative thoughts. This VR system allows users to transform their negative thoughts into virtual objects that can be manipulated, fostering mental detachment from these thoughts. The application's effectiveness was evaluated through a user study involving 20 participants, comparing the VR experience to a traditional mental imagery approach.

Here, I summarize several limitations of the paper that the authors need to address as outlined in their rebuttal.

1. A significant practical issue is the time-consuming and skill-dependent process of creating individualized 3D models based on participants' descriptions. While the authors suggest generative AI as a solution, this aspect requires more detailed discussion and exploration.
2. The study's sample size is relatively small, and the demographic was narrow, potentially limiting the generalizability of the findings. A broader and more diverse participant pool would strengthen the study's validity.
3. More comprehensive details about the equipment and setup used for the VR system are needed to ensure transparency and reproducibility.
4. Some reviewers noted a perceived lack of novelty in the study, comparing it to existing VR gaming experiences. Clarifying the unique aspects of this application and its differentiation from similar technologies would be beneficial.
5. The potential impact of participants' prior familiarity with VR on the study's results is not addressed. This factor should be considered in future studies.

The paper is well-written and presents a creative and promising application of VR in cognitive therapy. However, several limitations, particularly regarding the creation of virtual objects, sample size, and the need for more technical details, suggest that further refinement is needed. Given these considerations, the recommendation is a weak acceptance, acknowledging the potential of the work but also highlighting the need to address the identified limitations in future iterations.